# SERINC5 Restricts HIV-1 Infectivity by Promoting Conformational Changes and Accelerating Functional Inactivation of Env

**DOI:** 10.3390/v14071388

**Published:** 2022-06-25

**Authors:** Junghwa Kirschman, Mariana Marin, Yen-Cheng Chen, Junhua Chen, Alon Herschhorn, Amos B. Smith, Gregory B. Melikyan

**Affiliations:** 1Department of Pediatrics, Emory University, Atlanta, GA 30322, USA; junghwa.kirschman@gilead.com (J.K.); mmarin@emory.edu (M.M.); ychen71@emory.edu (Y.-C.C.); 2Children’s Healthcare of Atlanta, Atlanta, GA 30322, USA; 3Department of Chemistry, University of Pennsylvania, Philadelphia, PA 19104, USA; junhua.chen@jstar-research.com (J.C.); smithab@sas.upenn.edu (A.B.S.III); 4Division of Infectious Diseases and International Medicine, Department of Medicine, University of Minnesota, Minneapolis, MN 55455, USA; aherschh@umn.edu

**Keywords:** HIV-1, SERINC5, envelope glycoprotein inactivation, CD4 mimetic, fusion inhibitor

## Abstract

SERINC5 incorporates into HIV-1 particles and inhibits the ability of Env glycoprotein to mediate virus-cell fusion. SERINC5-resistance maps to Env, with primary isolates generally showing greater resistance than laboratory-adapted strains. Here, we examined a relationship between the inhibition of HIV-1 infectivity and the rate of Env inactivation using a panel of SERINC5-resistant and -sensitive HIV-1 Envs. SERINC5 incorporation into pseudoviruses resulted in a faster inactivation of sensitive compared to resistant Env strains. A correlation between fold reduction in infectivity and the rate of inactivation was also observed for multiple Env mutants known to stabilize and destabilize the closed Env structure. Unexpectedly, most mutations disfavoring the closed Env conformation rendered HIV-1 less sensitive to SERINC5. In contrast, functional inactivation of SERINC5-containing viruses was significantly accelerated in the presence of a CD4-mimetic compound, suggesting that CD4 binding sensitizes Env to SERINC5. Using a small molecule inhibitor that selectively targets the closed Env structure, we found that, surprisingly, SERINC5 increases the potency of this compound against a laboratory-adapted Env which prefers a partially open conformation, indicating that SERINC5 may stabilize the closed trimeric Env structure. Our results reveal a complex effect of SERINC5 on Env conformational dynamics that promotes Env inactivation and is likely responsible for the observed restriction phenotype.

## 1. Introduction

Serine incorporator 3 and 5 (SERINC3 and SERINC5) proteins inhibit retrovirus infectivity upon incorporation into virions [1,2], whereas another member of the SERINC family, SERINC2, has no antiviral activity [3,4,5]. Importantly, SERINC5-mediated restriction is antagonized by retro/lentiviral accessory proteins Nef, glycoGag and S2 that bind SERINC5 (hereafter abbreviated as SER5) and remove it from the plasma membrane, thereby reducing its incorporation into nascent retroviral particles [1,2,6,7,8]. The cellular functions of SERINCs have not been defined. It was originally concluded that SERINC proteins are involved in phospholipid biosynthesis [9], but lipidomic studies did not detect changes in the lipid compositions of SER5-expressing cells or SER5-containing viruses [10]. Structural analysis revealed that SER5 is a 10-transmembrane protein that contains a lipid binding pocket, but the identity of bound lipid(s) has not been determined [11].

Using single virus tracking, we have shown that SER5 blocks HIV-1 fusion at a stage prior to formation of small fusion pores [3], although cryo-electron tomography studies of virus–membrane vesicle fusion products suggests a block at a small pore dilation stage [12]. The mechanism of SER5-mediated restriction is not well understood and appears multifaceted. HIV-1 resistance to SER5 restriction maps to Env, specifically to the variable V1-V3 loops of gp120 [13,14]. Unliganded Env is known to sample multiple conformations, including the native (closed) conformation, and at least two distinct partially/fully open structures stabilized upon CD4 binding [15,16,17]. The V1-V3 loops located at the apex of Env trimer contribute to the Env’s stability and to the probability of finding the Env in the closed conformation [18,19,20]. Functional studies of Env mutants and chimeras suggest that SER5 resistance generally correlates with the preference of Envs of primary isolates for a closed conformation [2,3,13,14,21]. Accordingly, CD4 binding sensitizes the otherwise resistant AD8 Env to SER5-mediated restriction [21].

While SER5 may preferentially target the open conformation of HIV-1 Env, it also changes the Env conformation on virions, as evidenced by the altered antibody binding and neutralization sensitivity [3,5,11,13,22]. In addition, bimolecular fluorescence complementation (BiFC) assay for protein–protein interactions and virus antibody capture assays indicate that SER5 disrupts Env trimers [21,23]. Our recent super-resolution imaging work provided evidence for SER5-mediated disruption of the Env clusters on virions [4], which have been proposed to be important for viruses’ fusion competence [24]. Whether these SER5 effects on Env structure and distribution in the viral membrane are a result of direct interactions between these two proteins is currently unclear. Although Env–SER5 interaction has been reported using co-IP and BiFC assays [21], we were unable to co-IP these two proteins [3]. Moreover, using super-resolution microscopy, we found that SER5 does not colocalize with Env on virions [4]. The lack of colocalization is consistent with an indirect mechanism for SER5 antiviral activity.

To gain insights into the mechanism of SER5 antiviral activity, we followed up on our previous observation that SER5-mediated changes in Env conformation were associated with an accelerated loss of function of the sensitive HXB2 Env, but not resistant JRFL Env [3]. Here, the link between an accelerated loss of the Env function and SER5 restriction was investigated by examining a panel of resistant and sensitive HIV-1 Env glycoproteins and their mutants. SER5 sensitivity correlated well with faster Env inactivation in the presence of SER5, implying that accelerated loss of Env function underlies SER5-mediated restriction. Whereas Env mutations that favored a more open conformation reduced the efficiency of SER-mediated restriction, a CD4-mimetic compound was found to accelerate the loss of Env function on SER5-containing virions. This result suggest that CD4-bound Env is sensitized to SER5-mediated restriction. We further investigated the link between the native conformation of Env and SER5-sensitivity by probing the efficiency of the small molecule HIV-1 fusion inhibitor **484** [25], which preferentially recognizes the closed conformation of Env [25]. Surprisingly, virion-incorporated SER5 tends to sensitize the open conformation of a laboratory-adapted Env to inhibition by **484**, suggesting that this restriction factor induces a conformational state that is either more closed or can better interact with **484**.

## 2. Materials and Methods

### 2.1. Cell Lines, Plasmids, and Reagents

HEK293T/17 cells were purchased from ATCC (Manassas, VA), TZM-bl (donated by Drs. J.C. Kappes and X. Wu [26] and Cf2Th synCCR5 (referred to as Cf2/CCR5) (contributed by Drs. Tajib Mirzabekov and Joseph Sodroski [27]) cells were obtained through the NIH HIV Reagent Program, Division of AIDS, NIAID, NIH. The cells were passaged in high-glucose Dulbecco’s Modified Eagle’s Medium (Cellgro, Mediatech, Manassas, VA, USA) supplemented with 10% heat-inactivated fetal bovine serum (FBS, Atlanta Biologicals, Flowery Branch, GA, USA) and 100 units/mL penicillin/streptomycin (Gemini Bio-Products, Sacramento, CA, USA). The culture medium for HEK293T/17 cells included 0.5 mg/mL G418 (Cellgro).

The pCAGGS plasmids encoding HXB2 and JRFL envelope glycoproteins were provided by Dr. J. Binley (BioMed Institute, San Diego, CA, USA). pcRev (Dr. Bryan R. Cullen), NL4-3-E-R-Luc, pMM310-BlaM-Vpr, and pMDG-VSV-G expression plasmids were from the NIH HIV Reagent Program. GFP-Vpr expression plasmid was a gift from Dr. T. Hope (Northwestern University, Evanston, IL, USA). The HIV-1-based packaging vector pR9ΔEnvΔNef was a gift from Dr. Chris Aiken (Vanderbilt University, Nashville, TN, USA). PBJ5, PBJ5-SER5-HA, PBJ5-SER2-HA, CMV-SER5-iHA and CMV-SER2-iHA expression plasmids were described previously [3,4]. The ADA, ADA V1-alt [28], ADA N302Y, ADA R315Q [29], JR2, JR2 K683Q, JR2 F673L [30], and Comb-mut Env expressing plasmids were gifts from Dr. Michael Zwick (Scripps Research Institute, La Jolla, CA, USA). The pSVIII AD8, pSVIII AD8 S375W [31], pSVIII AD8 H66N [32], and HXB2 3.2P [33] were kindly provided by Dr. Hillel Haim (University of Iowa, Iowa City, IA, USA). JRFL I423A expression plasmid [34] was a gift from Dr. Joseph Sodroski (Dana-Farber Cancer Institute, Boston, MA, USA).

Bright-Glo luciferase kit and poly-D-lysine were purchased from Promega (Madison, WI, USA) and Sigma (St. Louis, MO, USA), respectively. The **484** and CD4-mimetic (BNM-III-170) compounds have been previously described [25,34,35]. Human 2G12 antibody was obtained from the NIH HIV Reagent Program. The antihuman AlexaFluor-647 and mouse anti-HA.11 were acquired from Invitrogen (Waltham, MA, USA) and BioLegend (San Diego, CA, USA), respectively. The anti-mouse CF568-conjugated was purchased from Biotium (Fremont, CA, USA).

### 2.2. Pseudovirus Production and Characterization 

To produce HIV-1 pseudoviruses, ~75% confluent HEK293T/17 cells grown in a 100-mm dish were transfected with 3 μg of Env expression plasmid, 4 μg of pR9ΔEnvΔNef, 1.5 μg of pBJ5 empty vector, pBJ5-SER5-HA or pBJ5-SER2-HA, 2 µg of BlaM-Vpr, and 1 μg of pcRev. For the experiments involving Cf2/CCR5 cells and immunostaining of pseudoviruses, the DNA transfection mix contained 3 μg Env expression plasmid, 6 μg of NL4-3-E-R-Luc, 0.6 μg of pcDNA3.1 empty vector or CMV-SER5-iHA or CMV-SER2-iHA, 2 µg of GFP-Vpr, and 1 μg of pcRev. Transfection was performed using JetPrime transfection reagent (Polyplus, Illkirch-Graffenstaden, France). After ~16 h, the supernatant of cells was replaced with phenol-red-negative growth medium (Life Technologies, Grand Island, NY, USA), and at 48 h post-transfection, the virus-containing supernatant was harvested and passed through a filter with 0.45µm PES membrane (VWR). Lenti-X concentrator (Clontech Laboratories, Mountain View, CA, USA) was used for further concentration of the virus. Virus stocks were stored at −80 °C. The p24/Gag amount of virus was measured by enzyme-linked immunosorbent assay (ELISA). Mouse anti-p24 monoclonal CA-183 (NIH HIV Reagent Program) was used for coating 96-well ELISA plates (Thermo Fisher Nunc, Rochester, NY, USA), and the capture ELISA was performed as previously described [36].

Pseudovirus immunostaining was performed as previously described [4]. Briefly, pseudoviruses were attached to 8-well glass-chambered coverslips (#1.5, Lab-Tek, Nalge Nunc International, Penfield, NY, USA) coated with poly-D-Lysine (0.1 mg/mL) by incubation at 4 °C for 30 min. Samples were fixed with 2% paraformaldehyde, blocked with 15% FBS/PBS^++^, and incubated with primary antibodies (5 μg/mL 2G12 for Env staining and mouse anti-HA.11 for SER-iHA staining) at 4 °C overnight. After 1 h incubation at room temperature with the second antibodies, samples were washed and imaged on a wide-field DeltaVision Elite microscope (Applied Precision, GE, Pittsburgh, PA, USA), using 40x oil objective (Olympus, Tokyo, Japan). 

### 2.3. Infectivity Assay 

For all infectivity assays, the target cells were plated a day before the experiment at 0.2 × 10^5^ cells/well in the 96-well black clear-bottom tissue-culture-treated plates (Greiner Bio-One North America Inc., Monroe, NC, USA). Pseudoviruses were bound to target cells by centrifugation at 4 °C for 30 min at 1550× *g*, and samples were incubated at 37 °C, 5% CO_2_ for 48 h. The samples were lysed with equal volumes of Bright-Glo luciferase substrate, and the activity of the reporter firefly luciferase was measured with a TopCount NXT plate reader (PerkinElmer Life Sciences, Waltham, MA, USA).

For measuring the rate of infectivity decay, 0.2 ng p24/well pseudoviruses in growth medium buffered with 20 mM HEPES were incubated for 0, 3, 8, and 24 h at 37 °C, 5% CO_2_ prior to inoculation of TZM-bl cells. In experiments using Cf2/CCR5 target cells, 20 µM BNM-III-170 was added to all samples prior to inoculation of the cells. For JRFL pseudovirus inactivation kinetics in the presence of CD4mimetic, pseudoviruses normalized for p24 content, as above, were incubated with 20 µM BNM-III-170 for 0, 45, 90, and 180 min at 37 °C, 5% CO_2_ prior to binding to Cf2/CCR5 target cells. For **484** IC_50_ measurements, the mixture of pseudovirus and **484** compound was incubated for 1 h at 37 °C, 5% CO_2_ prior to inoculation of TZM-bl cells.

### 2.4. Data Processing and Statistical Analysis

Curve-fitting of the experimental results, unpaired Student’s *t*-test, and two-way ANOVA repeated measures statistical analyses were performed using GraphPad Prism version 9.3.1 for Windows (GraphPad Software, La Jolla CA, USA). For T_50_ calculation, curve-fitting was performed using a single exponential decay (one-phase decay) equation [Y = (Y0 − Plateau) * exp(−K * X) + Plateau], where X is the time. For IC_50_, the curve-fitting was performed using [inhibitor] vs. normalized response (single site binding equation): [Y = 100/(1 + X/IC_50_)].

For immunostaining assay, an in-house-written Matlab protocol was used to identify single-GFP-Vpr-labeled virus particles by finding local maxima with Fast 2D peak finder (https://www.mathworks.com/matlabcentral/fileexchange/37388-fast-2d-peak-finder). To remove dim virus signals, a signal-to-background ratio smaller than 1.2 was implemented. The coordinates of GFP-Vpr were used to quantify the Env and SERINC fluorescence signals associated with these particles.

## 3. Results

### 3.1. HIV-1 Sensitivity to SER5 Restriction Correlates with the Rate of Spontaneous Loss of Infectivity

HIV-1 infectivity is known to decease over time under physiological conditions due to spontaneous functional inactivation of Env and the characteristic time of inactivation varies between different strains (e.g., [37]). The structural basis of functional Env inactivation is not understood, but it has been shown that inactivation is not strictly linked to a loss of the trimeric Env structure [37]. We have previously observed that SER5 incorporation accelerates functional inactivation of a sensitive HXB2 Env strain [3]. To generalize this finding, we examined a panel of SER5-sensitive and -resistant HIV-1 strains. VSV-G-pseudotyped viruses that are resistant to SER5 restriction [1,2] were used as a negative control. Pseudoviruses produced by transfection of HEK293T cells were collected and incubated for varied durations at 37 °C prior to infecting the indicator TZM-bl cells, and the resulting luciferase signal was measured after 48 h (Figure 1A), as previously described [3]. The half-time of Env inactivation (T_50_) was calculated from fitting the infectivity decay results with an exponential function (see Methods). The luciferase-based single-cycle infectivity assay has an excellent dynamic range (2.5–3 logs), allowing reliable measurements of infectivity decay over time.

In agreement with our published results, SER5 incorporation markedly accelerated the loss of infectivity of sensitive HXB2 pseudovirus, although the infectivity loss of another sensitive Env, ADA, was less pronounced (Figure 1B–D and Appendix A). A modestly accelerated loss of infectivity was detected for all tested SER5-resistant Env strains (Figure 1B–D and Appendix A). Control viruses containing SER2 did not exhibit significant increase in the rate of infectivity loss (Appendix A). Importantly, the Env sensitivity to SER5 (fold restriction) correlated well with the rate of functional inactivation (T_50_) of control viruses (Figure 1E), suggesting that the intrinsic Env stability is a good predictor of SER5 resistance. We further examined the link between inhibition of infectivity and the effect of SER5 on the rate of Env inactivation by plotting the ratio of T_50_ for control and SER5-containing viruses (fold-acceleration) as a function of fold-restriction by SER5 (Figure 1F). A correlation between the accelerated Env inactivation and reduction in infectivity implies that SER5 more potently accelerates functional inactivation of less stable Envs. Thus, the accelerated functional inactivation SER5-sensitive Envs may be responsible for the HIV-1 restriction phenotype.

### 3.2. Mutations Disfavoring the Closed Conformation of Env Do Not Generally Sensitize HIV-1 to SER5 Restriction

HIV-1 Env samples at least 3 distinct conformations revealed by single-molecule FRET measurements [15,16,17]. Laboratory adapted HIV-1 strains tend to spend more time in a partially opened asymmetric conformation that is favored upon CD4 binding, whereas primary isolates, like JRFL and AD8, which are generally more resistant to SER5 (Figure 1 and [1,2,3,13,21]), tend to maintain the native, closed conformation. We therefore asked if mutations in primary Envs that favor a more open Env conformation (hereafter referred to as open-conformation-promoting mutations) increase the sensitivity to SER5 restriction and, conversely, whether mutations that stabilize the closed conformation of Envs can confer resistance to this restriction factor. Toward this goal, we assembled a panel of open-conformation-promoting mutations in different resistant Env strains, as well as mutations that stabilize SER5-sensitive Env strains by increasing their thermal stability and/or resistance to conformation-specific neutralizing antibodies (Appendix A). The following Env open-conformation-promoting mutations were examined. The JRFL I423A mutation favors an open Env conformation [34], as do the JR2 gp41 MPER open-conformation-promoting mutations F673L and K683Q [30]. In the AD8 S375W mutant, the CD4 binding cavity is occupied by the bulky Trp side chain, creating a CD4-bound-like Env conformation [31] that is sensitive to spontaneous inactivation in the cold [38]. The following Env stabilizing mutations were tested. The HXB2 3.2P variant, which has been isolated from the rhesus macaques after infection with a SHIV construct, has multiple mutations in the variable regions of gp120 and gp41 and is resistant to neutralizing antibodies and cold inactivation [33]. The ADA V1-alt and comb-mutants contain multiple mutations that stabilize the native conformation of Env and confer resistance to heat inactivation and neutralization [28]. The N302Y and R315Q mutations stabilize the gp120 V3 region of ADA Env and increase its thermal stability [29]. The H66N mutation in AD8 Env reduces the sampling of a CD4-bound conformation and stabilizes against cold inactivation [32,38].

We compared fold-restriction of infection by SER5 and the kinetics of functional inactivation (T_50_) for the above panel of wild-type (WT) and mutant Env glycoproteins. The potency of SER5-mediated reduction of infectivity correlated well with and the intrinsic Env stability (T_50_) on control (vector) viruses (Figure 2A). Furthermore, the fold-acceleration of Env inactivation by SER5 also correlated with the Env sensitivity to this restriction factor (Appendix A). In other words, SER5 tends to more potently accelerate inactivation of less stable Envs that are susceptible to restriction. This correlation supports the role of the accelerated loss of Env function in SER5 restriction. Since most mutations were introduced in distinct sensitive and resistant Env background, the general effect of mutations stabilizing and disfavoring the close d conformation was assessed by plotting the fold acceleration of SER5-mediated inactivation of mutant relative to WT Env as a function of normalized fold restriction (ratio of mutant restriction over WT restriction, see the formulas in Figure 2B). This plot places all WT Env glycoproteins at the position 1/1 and shows the shifts in SER5 sensitivity and the relative rate of inactivation for all mutant Env glycoproteins (Figure 2B and Appendix AB,C). Here, a rightward and upward shift relative to WT represents increased fold restriction and fold acceleration of inactivation, respectively (blue arrows along the axes in Figure 2B). All stabilizing Env mutations rendered Env less sensitive to SER5 restriction, although not all changes reached statistical significance (Figure 2B, black symbols). Surprisingly, all open-conformation-promoting mutations either did not affect SER5 sensitivity or conferred a different degree of resistance to SER5 (Figure 2B, pink symbols). These results show that the open-conformation-promoting mutations examined here do not generally sensitize the virus to SER5 restriction or accelerate Env inactivation.

### 3.3. CD4-Bound Env Conformation Is More Sensitive to SER5 Restriction

Having observed a lack of correlation between SER5 resistance and mutations that stabilize or destabilize the closed Env conformation (Figure 2B), we asked if CD4 binding confers SER5 sensitivity—as has been proposed previously [21]—and accelerates Env inactivation. To test if CD4 binding modulates the Env resistance to SER5, we used a small-molecule CD4 mimetic (CD4mc) BNM-III-170, which has been shown to promote functionally relevant conformational changes in Env, enabling its interaction with coreceptors and subsequent induction of membrane fusion [39,40]. Since CD4mc competes with CD4 expressed on target cells and thereby inhibits HIV-1 infection [39,40], we used Cf2/CCR5 cells expressing the cognate coreceptor for JRFL Env but lacking CD4 [27]. Addition of CD4mc to JRFL pseudoviruses at the time of inoculation of Cf2/CCR5 cells resulted in efficient infection (Figure 3A). Preincubation of pseudoviruses at 37 °C without CD4mc followed by infection of Cf2/CCR5 cells in the presence of CD4mc revealed similar rates of spontaneous Env inactivation over time irrespective of SER5 or SER2 incorporation (Figure 3B), as expected for the SER5-resistant Env. To assess the effect of CD4mc on the kinetics of JRFL inactivation, JRFL-Env-pseudotyped particles were preincubated with 20 µM of CD4mc at 37 °C for varied durations, and the mixture was added to Cf2/CCR5 cells (Figure 3C). As expected, CD4mc markedly increased the rate of JRFL Env inactivation, irrespective of SER5 incorporation (Figure 3B,D). Control pseudoviruses lost infectivity more than sixfold faster in the presence of CD4mc compared to DMSO control (tables in Figure 3B,D). Importantly, upon binding to CD4mc, SER5-containing pseudoviruses were inactivated faster than control and SER2-containing pseudoviruses (Figure 3D). These findings support the notion that CD4 binding sensitizes HIV-1 Env to SER5-mediated restriction.

### 3.4. SER5 Sensitizes Env to Small Molecule Inhibitor Targeting the Closed Env

We have previously shown that SER5 increases the exposure of the MPER and heptad repeat 1 domains of gp41 [3]. In contrast, no significant differences in sensitivity to neutralization by anti-gp120 antibodies were detected. Having established that SER5 preferentially targets a CD4-bound conformation of JRFL Env, we asked if this restriction factor also promotes a more open conformation of the trimeric Env apex region. Herschhorn et al. have identified a small molecule inhibitor of HIV-1 fusion, 18A, that shows a strong preference for the closed conformation of Env and more potently inhibits primary isolates compared to some laboratory-adapted strains that prefer a more open conformation [25,34]. The compound **484** contains the N,N′-difunctionalized piperazine that is present in the entry inhibitor BMS-806 and shares several functional groups and mode of inhibition with 18A. We used the potency (IC_50_) of **484**, which is a more potent inhibitor than 18A, as an indicator of a “closeness” of the Env apex on control and SER5-containing viruses. [25]. Dose-response measurements confirmed that **484** inhibits the primary JR2 Env much more potently than the laboratory-adapted, CXCR4-tropic HXB2 Env (Figure 4A,B and Appendix A). SER5 incorporation into JR2 pseudoviruses did not significantly alter the potency of **484** (Figure 4A). In contrast, SER5—but not SER2, which incorporated into virions at levels similar to SER5 (Appendix A)—rendered HXB2 Env significantly more sensitive to **484** (Figure 4B). Of note, an excellent dynamic range of the luciferase-based infectivity assay allowed reliable infectivity measurements for control and SER5-containing viruses in the presence of high concentrations of **484** (Appendix A). The more potent inhibition of HXB2/SER5 pseudoviruses by **484** shows that, unexpectedly, SER5 facilitates HIV-1 neutralization by a compound that favors a more closed conformation of the trimeric Env apex region.

We then repeated the **484** dose–response measurements using other laboratory-adapted and primary isolates containing or lacking SER5 (Appendix A). A correlation between fold restriction of infectivity by SER5 and fold decrease in the IC_50_ of **484** (Figure 4C) supports the notion that SER5 synergizes with a compound that preferentially neutralizes a closed Env conformation, whereas resistant Env glycoproteins (except for AD8) do not exhibit noticeable change in IC_50_.

## 4. Discussion

Here, we examined the relationship between HIV-1 Env sensitivity to SER5-mediated restriction (fold decrease in infectivity) and the rate of spontaneous loss of Env function using a panel of Env strains and mutants. In agreement with our previous work showing a linked between these parameters for two HIV-1 strains (HXB2 and JRFL), we observed a good correlation between SER5-mediated reduction in infectivity and the half-time (T_50_) of Env inactivation for the tested primary and laboratory adapted HIV-1 isolates (Figure 1E). This correlation was also detected for multiple mutants shown to disrupt or stabilize the native, closed structure of Env (Figure 2A). Moreover, we observed a good correlation between the accelerated rates of Env inactivation on SER5 viruses and decrease in HIV-1 infectivity (Figure 1F, Figure 2B and Appendix A). Together, these results imply that (i) the intrinsic stability of Env is a predictor of its resistance to SER5 and (ii) an accelerated loss of HIV-1 Env function on SER5 viruses is a major cause of decreased infectivity. These viruses may lose infectivity at physiological conditions, before they attach to target cells. We therefore predict that cell–cell HIV-1 transmission, which is a much faster and more efficient mode of transmission than cell–free virus infection, would be less susceptible to SER5 restriction.

Previous work has suggested that CD4-bound Env conformation is more susceptible to SER5-mediated inhibition [21]. However, this study explored the effect of CD4 and SER5 co-expression in virus producing cells where the Env-CD4 interactions cannot be controlled. To shift Env to a CD4-bound conformation under more tractable conditions, we pretreated pseudoviruses with CD4mc and measured the rate of infectivity loss in control and SER5-containing viruses (Figure 3). These experiments showed that, indeed, CD4 binding sensitizes the otherwise resistant JRFL Env to SER5-mediated restriction, in agreement with the previous report [41]. Importantly, however, our attempts to sensitize Env by introducing mutations that favor an open, CD4-bound conformation were not successful. Whereas a correlation between fold restriction and the rate of infectivity inactivation was maintained across mutations that both stabilize and destabilize the closed Env conformation, some mutations favoring a more open Env conformation actually reduced the efficiency of SER5-mediated restriction Figure 2B and Appendix A). This finding suggests that the CD4-bound-like Env conformations favored by this set of mutants are not selectively targeted by SER5; however, SER5 shows preference for the actual CD4-bound conformation. Since soluble CD4 binding itself promotes Env inactivation ([42,43] and Figure 3), it is possible that this destabilizing effect synergizes with SER5-mediated destabilization to greatly accelerate loss of infectivity.

Aside from affecting a CD4-bound Env conformation, SER5 itself appears to alter the Env structure. We and others have previously shown that SER5 incorporation into virions induces or stabilizes conformations that are consistent with a more open Env structure [3,5,13]. Specifically, SER5 increased the exposure of the gp41 MPER and heptad repeat 1 (HR1) domains on both resistant and sensitive Envs [3]. However, we were unable to detect SER5-induced antigenic changes in the gp120 subunit (but see [23] for a different conclusion). To study possible conformational changes in the trimeric Env apex, which is largely responsible for SER5 resistance, we used a small-molecule inhibitor **484** which preferentially neutralizes the Env closed conformation [25,34]. Thus, **484** more potently inhibits primary Env strains that prefer the closed conformation, while exhibiting poor potency against laboratory-adapted strains that are mainly in an open conformation [25,34]. Surprisingly, we found that SER5 increases the potency of **484** against otherwise drug-resistant, CXCR4-tropic HXB2 Env (Figure 4), similar to the previous report of potentiation of **484** inhibition of another SER5-sensitive Env, SF162, which uses CCR5 as a coreceptor for HIV-1 entry [5]. This important result suggests that SER5 might promote a novel Env conformation in which the MPER and HR1 domains are more exposed, while, counterintuitively, the trimeric Env apex is sensitive to compounds that preferentially neutralize a closed-like conformation. Future structural studies of Env on SER5-containing virions should be able to test this prediction.

There is currently no consensus on whether SER5 directly interacts with HIV-1 Env. Whereas there is evidence for SER5–Env interaction [21], the lack of Env colocalization with SER5 on virions observed in our super-resolution imaging study [4] supports an indirect effect of SER5 on Env structure and function. A direct interaction model would require SER5 to recognize diverse retroviral Env glycoproteins having little sequence homology. Moreover, a recent work has demonstrated the ability of SER5 to restrict non-retroviral viral glycoproteins [44]. We thus propose that virus-incorporated SER5 exerts its antiviral activity indirectly by modifying the properties of viral membrane. Indeed, oxidation of HIV-1 membrane cholesterol disrupts Env clusters and inhibits infectivity [45]. It is thus possible that, through its lipid-binding properties [11], SER5 can alter the viral membrane structure and thereby inhibit the Env function.

## Figures and Tables

**Figure 1 viruses-14-01388-f001:**
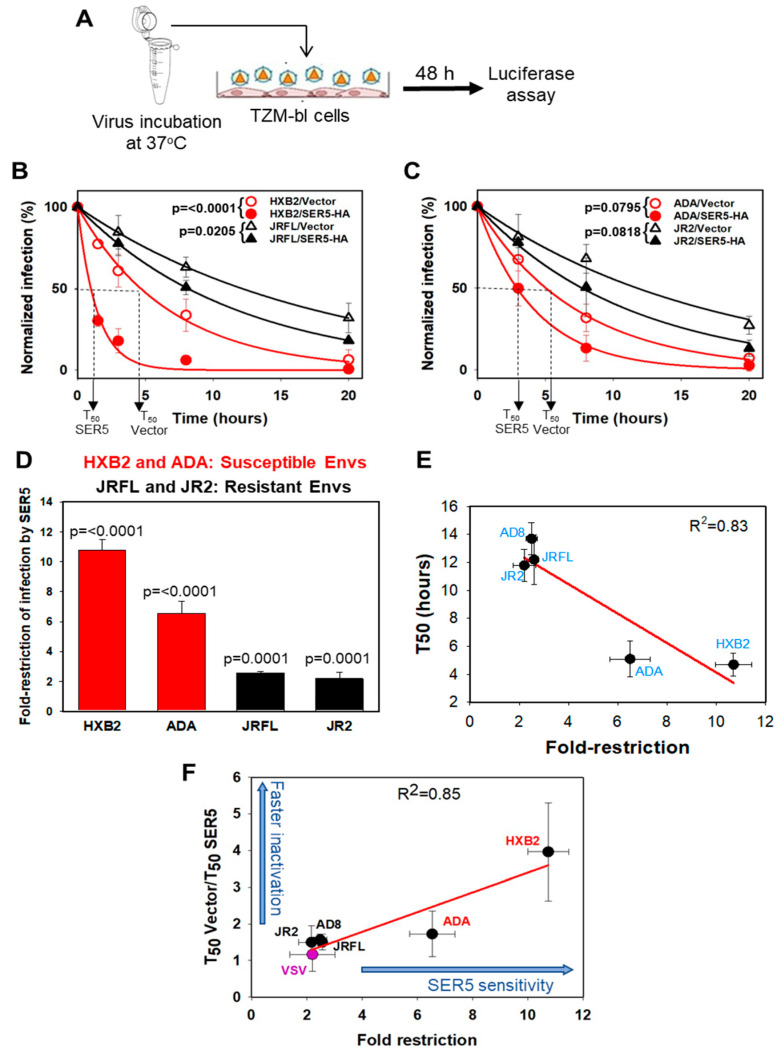
Spontaneous inactivation of HIV-1 Env glycoproteins induced by SER5 correlates with the degree of restriction. (**A**) Schematic diagram of experimental setup. Pseudoviruses were incubated at 37 °C, 5% CO_2_ in HEPES-buffered medium for various durations. Pseudovirus infectivity was monitored by firefly luciferase activity in TZM-bl cells at 48 h post-infection. (**B**,**C**) Rates of infectivity decay for sensitive (red curves) vs. more resistant HIV-1 Envs (black curves). HIV-1 pseudoviruses containing or lacking SER5-HA were incubated at 37 °C, 5% CO_2_, as indicated in (**A**). The luciferase signal for each pseudovirus was normalized to its respective freshly initiated preparation. Data points are means and S.D. from four (HXB2), and three (JRFL, ADA, and JR2) independent experiments, each performed in triplicate. The infectivity decay half-life (T_50_) values calculated by single exponential decay (one-phase decay) fitting are listed in Appendix A. The statistical significance of SER5-HA’s effect on HIV-1 rates of decay (ANOVA repeated measures test) is shown in the plots. (**D**) Fold restriction by SER5-HA of two sensitive (red), and two resistant (black) HIV-1 Envs analyzed in (**B**,**C**). Data are means and S.D. from the independent experiments mentioned in (**B**,**C**). The statistical significance analyzed by a two-tailed t-test is shown in the plot. (**E**) Correlation between the rate of Env inactivation (T_50_) on control (vector) viruses and fold-restriction by SER5 for selected sensitive and resistant Envs. (**F**) Correlation between SER5-HA fold-restriction and SER5-mediated acceleration of infectivity decay (T_50_Vector/T_50_SER5) for two sensitive Envs (red), and three resistant Envs (black). VSV-G Env (purple) was used as control. The red line is the linear regression. The blue arrows along the axes show shifts from the regression line corresponding to SER5 sensitization. The data in panels E and F are means and S.D. from two to four independent experiments, each in triplicate. The number of independent experiments for each pseudovirus together with the values of fold restriction and T_50_ ratio are presented in Appendix A.

**Figure 2 viruses-14-01388-f002:**
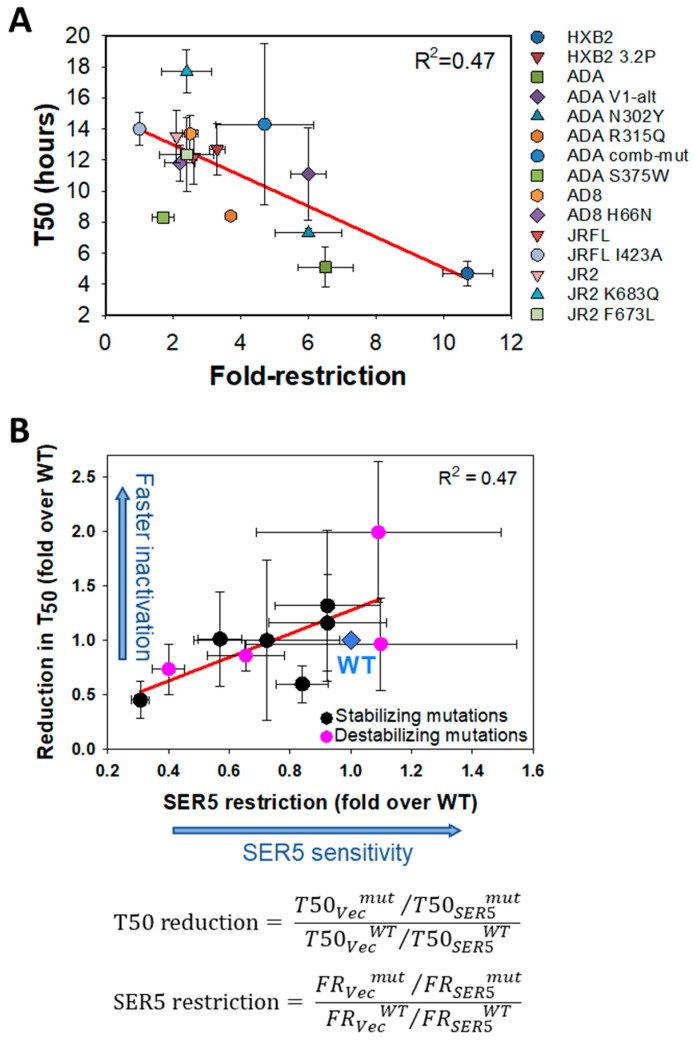
Correlation between the rates of inactivation and fold restriction by SER5 for wild-type and mutant HIV-1 Env glycoproteins. The experiments were performed as depicted in Figure 1A. (**A**) Scatter plot showing correlation between T_50_ and SER5 fold restriction for five wild-type (WT) Envs and 10 Env mutants. A linear regression plot is shown by the red line. The data are means and S.D. of two to four independent experiments, each performed in triplicate. The values of T_50_ ratio, fold restriction, and the numbers of independent experiments are listed in Appendix A. (**B**) Scatter plot showing correlation between SER5-mediated acceleration of Env inactivation relative to WT and SER5 restriction for six Envs containing stabilizing mutations (HXB2 3.2P, ADA N302Y, ADA V1 alt, ADA R315Q, ADA Comb-mutant, and AD8 H66N.—black circles), and four Envs containing open-conformation-promoting mutations: JRFL I432A, AD8 S375W, JR2 F673L, and JR2 K683Q—pink circles (referred to as destabilizing mutations on the graph). The WT values across different Env strains were set to 1 (blue symbol). A red line is the linear regression. Blue arrows along axis indicate faster Env inactivation and stronger SER5 restriction. The plotted values were calculated using the formulas shown below the graphs. The values used in calculations and the number of experiments are listed in Appendix A. Abbreviations used: *vec,* vector; *mut,* mutant; WT, wild-type; and FR, fold restriction.

**Figure 3 viruses-14-01388-f003:**
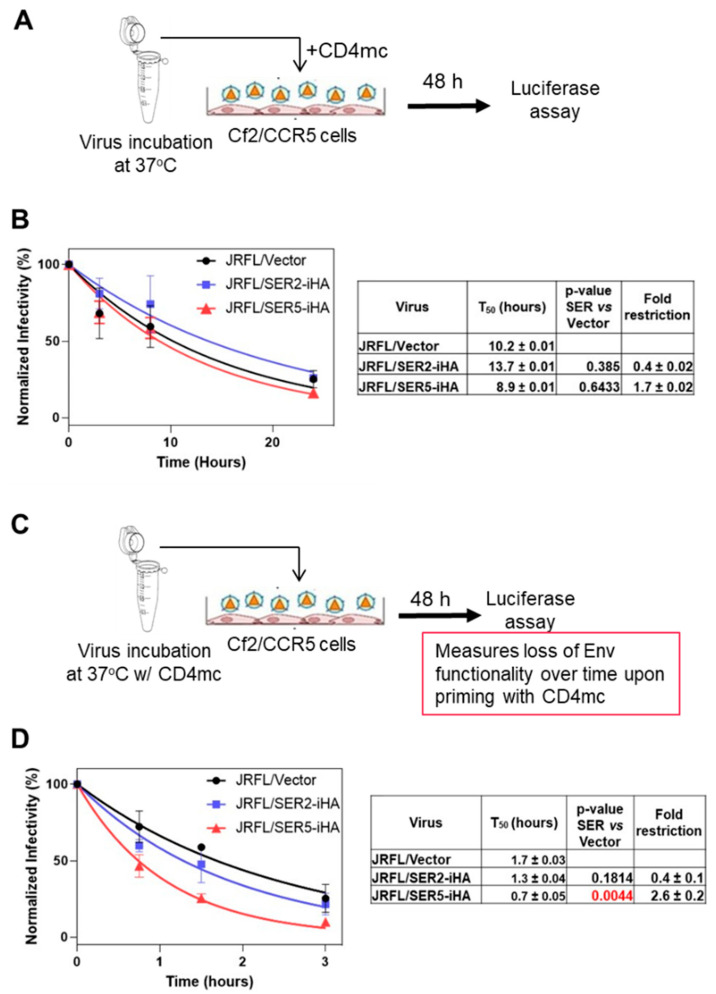
CD4 mimetic BNM-III-170 selectively accelerates the infectivity decay of HIV-1 JRFL pseudoviruses containing SER5. (**A**) A diagram of the rate of decay experiments related to panel B. Briefly, JRFL HIV-1 pseudoviruses lacking or containing SER5-iHA or SER2-iHA were incubated at 37 °C, 5% CO_2_ for various durations. Prior to infection of Cf2/CCR5 cells, 20 µM BNM-III-170 (CD4mc) was added to each sample, and the luciferase signal was measured 48 h post-infection. (**B**) The rates of infectivity decay of JRFL HIV-1 pseudoviruses lacking or containing SER5-iHA or SER2-iHA. The plotted data are means and S.D. from two independent experiments performed in triplicate. The T_50_ (half-life) values calculated by single-exponential decay fitting, statistical significance of SER-iHA effect on the JRFL rate of decay (ANOVA repeated measures test), and the fold restriction are listed in the table on the right. (**C**) A diagram of the assay to measure the Env inactivation kinetics in the presence of CD4mc. The JRFL pseudoviruses were incubated at 37 °C, 5% CO_2_ for various durations in the presence of 20 µM BNM-III-170 (CD4mc), followed by inoculation of Cf2/CCR5 target cells. The resulting infection was measured after 48 h wish a luciferase assay. (**D**) The graph shows the result of infectivity decay fitted with a single-exponential function. Data are means and S.D. of three independent experiments, each performed in triplicate. Table on the right lists the T_50_ values and statistical significance (ANOVA repeated measures test), as well as fold-restriction of JRFL virus by SER2-iHA and SER5-iHA.

**Figure 4 viruses-14-01388-f004:**
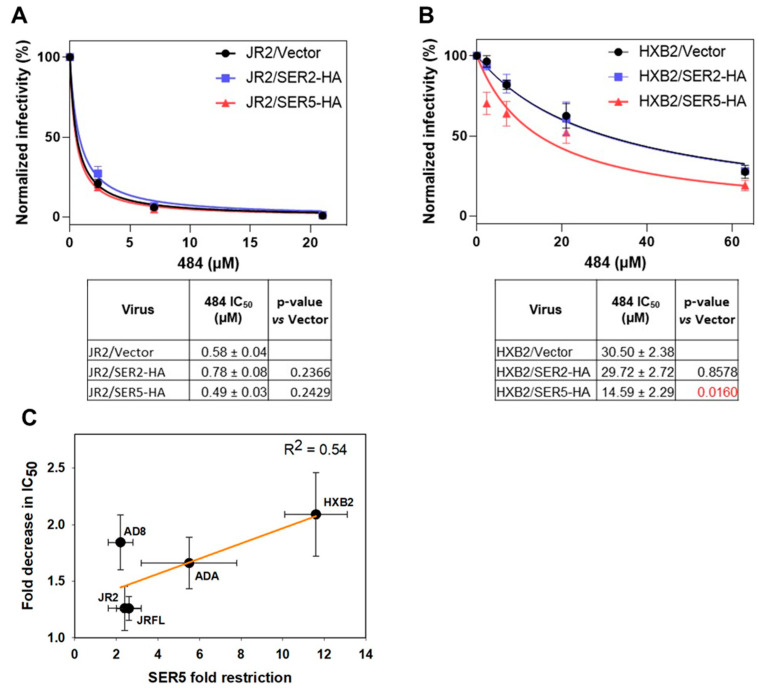
Virion-incorporated SER5 tends to sensitize HIV-1 Env glycoproteins to **484** inhibitor that is selective for the closed Env conformation. (**A**,**B**) HIV-1 pseudoviruses lacking or containing SER2-HA or SER5-HA were pre-treated at 37 °C, 5% CO_2_ with various concentrations of **484** compound for 1 h, followed by infection of TZM-bl cells. The infectivity was quantified after 48 h by luciferase assay. Graphs show the data and curve-fitting for JR2 (A) and HXB2 (**B**) pseudoviruses. Data are means and S.D. of two (JR2 pseudoviruses), three (HXB2/SER5-HA), and two (HXB2/SER2-HA) independent combined experiments, each performed in triplicate. The IC_50_ values obtained by curve-fitting and statistical significance (ANOVA repeated measures test) are shown in the tables below the graphs. (**C**) Scatter plot showing a correlation between fold decrease in **484** IC_50_ and fold restriction by SER5 for five HIV-1 Envs. The dark red line indicates linear regression. The IC_50_ values obtained by curve-fitting all studied Envs and statistical significance (ANOVA repeated measures test) are listed in Appendix A.

## Data Availability

Not applicable.

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
