# Peer review of "SERINC5 Restricts HIV-1 Infectivity by Promoting Conformational Changes and Accelerating Functional Inactivation of Env"

_viruses, 2022, doi:10.3390/v14071388_

Round 1
Reviewer 1 Report
In their very interesting study, Melikyan and colleagues found that the rate of spontaneous inactivation of HIV-1 strains correlates with their susceptibility to restriction by SERINC5. Moreover, SERINC5 appears to accelerate the rate of HIV-1 inactivation, as if the effect of SERINC5 was taking time to manifest. They also showed that a CD4-mimetic inhibitor sensitizes HIV-1 envelope to SERINC5 restriction: the JRFL isolate normally insensitive to SERINC5 inactivated faster in the presence of SERINC5. Overall, these data clarify some of the aspects of the SERINC5-Env "interaction" and will be helpful to eventually figure out the mechanism of HIV-1 restriction by SERINC5 and related proteins.
Minor comments:
1. In light of these data, it is very tempting to visualize HIV-1 restriction by SERINC5 as acceleration of spontaneous inactivation of Env. However, it may not be the case, and the structural bases for spontaneous inactivation of Env and restriction by SERINC5 may be entirely different. I suggest toning down this conclusion in lines 201-203. Furthermore, on the subject of spontaneous inactivation of Env: could the authors clarify what mechanism(s) may be involved in this process? Is anything known about the state of Env on senescent virions?
2. One would expect that "destabilizing" mutations to accelerate spontaneous virus inactivation, which is not the case (Fig. 2B). Are these mutations truly destabilizing? If they are not accelerating the infectivity loss, then it is not really surprising that they do not increase virus sensitivity to SERINC5 (given the correlation).
3. Experiment in Fig. 4B uses infectious virus (HXB2) and nearly dead virus (HXB2+SERINC5) and compares their sensitivity to a very weak compound (EC50~10 uM or more). I wonder if the small difference on normalized infectivity plots is real. Are we looking at a fraction of abnormal HXB2 Env trimers, which for some reason were not inactivated by SERINC5?
Author Response
Minor comments:
- In light of these data, it is very tempting to visualize HIV-1 restriction by SERINC5 as acceleration of spontaneous inactivation of Env. However, it may not be the case, and the structural bases for spontaneous inactivation of Env and restriction by SERINC5 may be entirely different. I suggest toning down this conclusion in lines 201-203. Furthermore, on the subject of spontaneous inactivation of Env: could the authors clarify what mechanism(s) may be involved in this process? Is anything known about the state of Env on senescent virions?
Response: We thank the reviewer for this comment and edited the sentence (lines 206-207 of the revised manuscript) as follows:
“Thus, the accelerated functional inactivation SER5-sensitive Envs may be responsible for the HIV-1 restriction phenotype.”
We talk about functional inactivation of Env without implying a specific structure(s). We merely show a correlation between the virus’ SER5 sensitivity and the rate of functional Env inactivation, which suggests causative relationship. The structural basis of Env inactivation is not well understood. An excellent study by Zwick’s lab has measured the inactivation kinetics for several Envs (doi:10.1371/journal.pone.0021339) and shown using blue-native gel that inactivated Env does not have to fully lose trimeric structure. We now added the following sentence to the first paragraph of Results:
“The structural basis of functional Env inactivation is not understood, but it has been shown that inactivation is not strictly linked to a loss of the trimeric Env structure [37].”
- One would expect that "destabilizing" mutations to accelerate spontaneous virus inactivation, which is not the case (Fig. 2B). Are these mutations truly destabilizing? If they are not accelerating the infectivity loss, then it is not really surprising that they do not increase virus sensitivity to SERINC5 (given the correlation).
Response: This is an excellent point. We referred to mutations that promote an open Env conformation as destabilizing, since they are known to destabilize the closed conformation. However, we see how this could cause confusion, given the lack of clear effect of these mutations on the rate of functional Env inactivation (Fig. 2A, Table 1). (Note that Fig. 2B does not plot the rates of WT and mutant inactivation; instead, it plots the normalized ratios of the effect of SER5 on WT vs mutant Env). Our results in Fig. 2B show that mutations that favor an open conformation(s) of Env do not sensitize the virus to SER5. For clarity, we now refer to these as “open conformation-promoting mutations” throughout the text.
- Experiment in Fig. 4B uses infectious virus (HXB2) and nearly dead virus (HXB2+SERINC5) and compares their sensitivity to a very weak compound (EC50~10 uM or more). I wonder if the small difference on normalized infectivity plots is real. Are we looking at a fraction of abnormal HXB2 Env trimers, which for some reason were not inactivated by SERINC5?
Response: The luciferase-based infectivity assay has a very large dynamics range (at least 2.5-logs), and we always make sure that our measurements are within the linear range, away from the background level or signal saturation. An ~11-fold reduction in HXB2 infectivity by SER5 (Fig. 4C, Table 1) is well within our dynamic range. The new Suppl. Fig. S3 (related to Fig. 4) shows that, even for SER5-containing JR2 and HXB2 pseudoviruses, raw infectivity values (luciferase signals) are more than an order of magnitude greater than the background. We also show (Suppl. Fig. S3C) that the JR2-generated luciferase signal at the highest 484 concentration is still several-fold greater than background signal. The effect of SER5 on 484 sensitivity is highly statistically significant and is therefore real. Besides, these measurements were performed for other, less sensitive Envs, and a correlation between fold-restriction and IC50 was observed (Fig. 4C). We now noted the dynamic range of our assay in the first paragraph of Results:
“The luciferase-based single-cycle infectivity assay has an excellent dynamic range (2.5-3 logs), allowing reliable measurements of infectivity decay over time.”
Regarding “abnormal” HXB2 trimers, HIV-1 Env is known to be heterogeneous, so it is possible that a sub-population of Envs is less susceptible to SER5 restriction. We are not sure, however, how this relates to the interpretation of our 484 titration results, as this population should be presented in both control and SER5-containing viruses. The dramatically higher resistance of HXB2 to 484 inhibition compared to JR2 is in excellent agreement with the literature (doi: 10.1038/nchembio.1623; doi: 10.1038/s41467-017-01119-w).
Reviewer 2 Report
The mechanism by which SER5 restricts retroviral infectivity remains obscure. This manuscript expands on a previous study from the same authors, who previously reported that SER5 causes functional inactivation of HIV-1 Env. In the present manuscript, the authors convincingly confirm a correlation between Env inactivation and SER5 restriction potency using a wider panel of Env isolates and mutants. They also investigate whether substitutions of Env supposed to affect the closed conformation of the Env glycoprotein alter sensitivity to SER5, without finding clear evidence in support of a correlation with the conformational state of Env.
The manuscript is well organized, and all necessary controls are present. The main outcome, which reinforces the notion that SER5 alters functional Env inactivation, is an important contribution toward deciphering the restriction mechanism. However, a caveat of the study is that all experiments are performed using a pseudotype system in which SER5 in producing cells is ectopically expressed. To ascertain the relevance of these results, it would be important to establish whether they can be reproduced with SER5 endogenously expressed. I therefore strongly suggest the addition of one experiment for the comparison of functional Env inactivation in virions produced in a cell line endogenously expressing SERINC5 and its SER5 KO counterpart.
Author Response
Reviewer 2 |
|
|
|
|
The mechanism by which SER5 restricts retroviral infectivity remains obscure. This manuscript expands on a previous study from the same authors, who previously reported that SER5 causes functional inactivation of HIV-1 Env. In the present manuscript, the authors convincingly confirm a correlation between Env inactivation and SER5 restriction potency using a wider panel of Env isolates and mutants. They also investigate whether substitutions of Env supposed to affect the closed conformation of the Env glycoprotein alter sensitivity to SER5, without finding clear evidence in support of a correlation with the conformational state of Env.
The manuscript is well organized, and all necessary controls are present. The main outcome, which reinforces the notion that SER5 alters functional Env inactivation, is an important contribution toward deciphering the restriction mechanism. However, a caveat of the study is that all experiments are performed using a pseudotype system in which SER5 in producing cells is ectopically expressed. To ascertain the relevance of these results, it would be important to establish whether they can be reproduced with SER5 endogenously expressed. I therefore strongly suggest the addition of one experiment for the comparison of functional Env inactivation in virions produced in a cell line endogenously expressing SERINC5 and its SER5 KO counterpart.
Response: We thank the reviewer for the positive comments and for a very good suggestion. We just would like to point out that we use the pBJ5 vector to express low levels of SER5 in HEK293T cells (Rosa et al., Nature 2015). CD4 T-cells endogenously express both SER3 and SER5, but, unfortunately, we are currently not set up to work with replication-competent HIV-1 and perform these experiments. We certainly plan to implement this assay to probe the effects of endogenous SER3/SER5 in the future.
Reviewer 3 Report
This study examined the molecular mechanisms by which SERINC5 restricts HIV-1 infectivity. Several complementary approaches were used, measuring the rate of inactivation of HIV-1 particles carrying SERINC5, examining HIV-1 Env stabilizing and destabilizing mutants, testing different HIV-1 Env-binding inhibitors including a CD4-mimetic compound and compound 484 which neutralizes closed Env. Interesting observations were made, provide new insights into how SERINC5 acts on HIV-1 Env and which types of HIV-1 Env is more susceptible to SERINC5 restriction. For example, CD4 binding, which opens HIV-1 Env, sensitizes Env to SERINC5, which is in agreement with what has been reported in the literature. Unexpectedly, SERINC5 increases the potency of compound 484 in inhibiting Env which tends to adopt open conformation.
Fig 1: how does SERINC5 accelerate spontaneous HIV-1 Env inactivation? Is Env shedding one of the causes? Can the authors measure this shedding effect?
Fig. 2: does T50 (inactivation rate) equal to stabilizing/destabilizing property, open/close Env conformation? A few concepts are used to describe these Envs.
Fig. 4: JR2 Env is much more sensitive to compound 484 than HXB2 Env, does this affect how SERINC5 affects the sensitivity of these two Envs to compound 484? Can there be a different interpretation of the data? What if lower concentrations of 484 are used to test JR2 Env to test SERINC5 in the 50% inhibition window?
Author Response
Reviewer 3
This study examined the molecular mechanisms by which SERINC5 restricts HIV-1 infectivity. Several complementary approaches were used, measuring the rate of inactivation of HIV-1 particles carrying SERINC5, examining HIV-1 Env stabilizing and destabilizing mutants, testing different HIV-1 Env-binding inhibitors including a CD4-mimetic compound and compound 484 which neutralizes closed Env. Interesting observations were made, provide new insights into how SERINC5 acts on HIV-1 Env and which types of HIV-1 Env is more susceptible to SERINC5 restriction. For example, CD4 binding, which opens HIV-1 Env, sensitizes Env to SERINC5, which is in agreement with what has been reported in the literature. Unexpectedly, SERINC5 increases the potency of compound 484 in inhibiting Env which tends to adopt open conformation.
Fig 1: how does SERINC5 accelerate spontaneous HIV-1 Env inactivation? Is Env shedding one of the causes? Can the authors measure this shedding effect?
Response: We thank the reviewer for the positive comments. The structural basis for spontaneous Env inactivation is not understood. Michael Zwick’s lab has shown that functionally inactivated Env can, at least partially, maintain the trimeric structure (doi:10.1371/journal.pone.0021339). This agrees with our published results that SER5 does not induce gp120 shedding (see Suppl. Fig. S2C in Sood et al., JBC 2017). We did not find evidence for direct Env-SER5 interactions (Sood et al., JBC 2017; Chen et al., ACS Nano 2020) and therefore believe that SER5 exerts its effect indirectly, through modifying the lipid organization of viral membrane. We currently have a manuscript under revision describing the effects of SER5 on the lipid order of the HIV-1 membrane.
Fig. 2: does T50 (inactivation rate) equal to stabilizing/destabilizing property, open/close Env conformation? A few concepts are used to describe these Envs.
Response: We agree with the reviewer that our terminology was somewhat confusing, as was also pointed out by Reviewer 1. We now refer to mutations that disfavor the open conformation of Env as “open conformation-promoting mutations”, instead of destabilizing mutations.
Fig. 4: JR2 Env is much more sensitive to compound 484 than HXB2 Env, does this affect how SERINC5 affects the sensitivity of these two Envs to compound 484? Can there be a different interpretation of the data? What if lower concentrations of 484 are used to test JR2 Env to test SERINC5 in the 50% inhibition window?
Response: We tested the 484 efficacies for a panel of Envs exhibiting different sensitivities to SER5 (Fig. 4C) and observed a good correlation, suggesting that SER5 tends make Env more sensitive to 484. We also performed an additional experiment for JR2 pseudoviruses using a lower concentration of 484 (see the figure below) and obtained a similar IC50 value to that in Fig. 4A. Additionally, removal of the lower concentration point, did not substantially affect the IC50 of 484 (compare the upper and lower tables on the right), supporting robustness of our measurements. Please note that, like for HXB2, SER5 tends to lower the IC50 value for the highly 484-sensitive JR2 and JRFL viruses, but this effect does not reach statistical significance (Fig. 4 and Table 2).
We interpreted these finding in accordance with structural, genetic, and functional data by Herschhorn and Sodroski that convincingly show that 484 (and its analog 18A) target the closed trimeric Env apex structure (doi: 10.1038/nchembio.1623; doi: 10.1038/s41467-017-01119-w). We also provided an alternative explanation for these results in the original manuscript (lines 86-87) by stating that “484 induces a conformational state that is either more closed or can better interact with 484”.

Round 2
Reviewer 2 Report
I am satisfied with the response of the reviewers.